# Newcastle Disease Virus as a Vaccine Vector for SARS-CoV-2

**DOI:** 10.3390/pathogens9080619

**Published:** 2020-07-29

**Authors:** Edris Shirvani, Siba K. Samal

**Affiliations:** Virginia-Maryland College of Veterinary Medicine, University of Maryland, College Park, MD 20742, USA; eshirvan@umd.edu

**Keywords:** SARS-CoV-2, COVID-19, coronavirus, Newcastle disease virus (NDV), vaccine, viral vector and viral respiratory infection

## Abstract

The emergence of severe acute respiratory syndrome coronavirus 2 (SARS-CoV-2) has resulted in more than 16 million infections and more than 600,000 deaths worldwide. There is an urgent need to develop a safe and effective vaccine against SARS-CoV-2. Currently, several strategies are being pursued to develop a safe and effective SARS-CoV-2 vaccine. However, each vaccine strategy has distinct advantages and disadvantages. Therefore, it is important to evaluate multiple vaccine platforms to select the most efficient vaccine platform for SARS-CoV-2. In this regard, Newcastle disease virus (NDV), an avian virus, has several well-suited properties for development of a vector vaccine against SARS-CoV-2. Here, we elaborate on the idea of considering NDV as a vaccine vector for SARS-CoV-2.

## 1. Introduction

The ongoing pandemic of coronavirus disease 2019 (COVID-19) has already resulted in more than 16 million infections and more than 600,000 deaths, and has devastated the livelihoods of people globally. The causative agent is a coronavirus that has sequence similarity with the severe acute respiratory syndrome (SARS) coronavirus of 2003 (SARS-CoV) [1,2,3,4]. This new virus is named SARS-CoV-2. Currently, no vaccine or antiviral is available against SARS-CoV-2. A safe and effective vaccine is urgently needed to control the current pandemic and any potential future pandemic.

Coronaviruses are enveloped, positive-sense, single-stranded RNA viruses belonging to the family *Coronaviridae.* These viruses mostly infect animals, including avian species and mammals. In humans, they generally cause mild respiratory infections [5]. However, some zoonotic coronaviruses, which include the SARS-CoV, Middle East respiratory syndrome coronavirus (MERS-CoV), and SARS-CoV-2, have caused severe respiratory disease in humans [1,6,7]. The genome of coronavirus is approximately 30 kb long and encodes several structural and nonstructural proteins. Proteins that form the structure of coronavirus are the spike (S), envelope (E), membrane (M), and nucleocapsid (N). The S gene with a size of 3.8 kb in length encodes the S protein. The S protein is the largest of all coronavirus structural proteins with a size of 128 kDa to 160 kDa. The S protein is heavily glycosylated and forms a crown-like structure on the envelope of the virion. The S protein mediates attachment and entry of the virion into the cell, and is the major protective antigen [5]. From studies of SARS, MERS, and avian coronaviruses, it is well known that a successful vaccine must incorporate the whole S protein [8,9,10]. The major neutralizing antigenic sites on this protein are highly conformation-dependent. Therefore, the S protein, to be effective as a vaccine antigen, must be presented to the host in its native conformation.

## 2. Why Use a Virus Vector Vaccine for SARS-CoV-2?

Currently, several different approaches are being pursued at a rapid speed to develop a safe and effective vaccine against SARS-CoV-2 [11]. Among these strategies, a viral vector vaccine offers a number of advantages over other vaccines.

First of all, it provides a live-virus vaccine approach that does not require involvement of the complete pathogen, which avoids the handling of live SARS-CoV-2. A live viral vector vaccine mimics the natural infection of the pathogen against which the vaccine is made. The vector infects cells in the target host and expresses the foreign antigen intracellularly, inducing both innate and adaptive immunity. The in vivo expression of the foreign antigen in target host cells presents the protective epitopes in native conformation, which is particularly important for the S protein of SARS-CoV-2. Replication of the vector in the host enhances the magnitude of the immune response and often requires a low dose of the vaccine. Some viral vectors are respirotropic, which are more suited for immunization against a respiratory pathogen like SARS-CoV-2 because they can induce local as well as systemic immunity. Another advantage of viral vector vaccines is that they can be produced rapidly and cheaply. Vector vaccines are not associated with the risk of reversion to virulence, a concern with live-attenuated vaccines, or with the risk of incomplete inactivation of inactivated vaccines. However, viral vector vaccines also have some challenges. Most viral vector vaccines are replication-competent in vivo; therefore, they must be non-pathogenic to the target host, and have sufficient replication and antigen expression to elicit a protective immune response. Sometimes, it is difficult to achieve both with a replication-competent vector. However, these challenges can be avoided by choosing a naturally attenuated virus vector that has a good track record of safety as a live vaccine in other species.

A viral vector vaccine for SARS-CoV-2 must be highly safe and capable of inducing a protective immune response. Currently, a number of DNA and RNA virus vector platforms are under evaluation for a SARS-CoV-2 vaccine, including attenuated vaccinia virus, replication-defective adenovirus, vesicular stomatitis virus, human parainfluenza viruses, and alphavirus replicons. However, each viral vector has some limitations that may or may not be possible to overcome. For example, the immunogenicity of some vaccinia virus vector vaccines has not been satisfactory, replication-defective adenovirus vector vaccines require a high dose and may not induce good local immunity, the safety of vesicular stomatitis virus vector vaccine in humans is questionable, human parainfluenza virus vector vaccines may not be effective in adults due to pre-existing immunity to the vector, and alphavirus vector vaccines are difficult to manufacture in large scale. Keeping these limitations in mind, we think Newcastle disease virus (NDV), as avian virus, has a number of characteristics that make it suitable for use as a vaccine vector for SARS-CoV-2.

## 3. Newcastle Disease Virus

NDV is an enveloped, negative-sense, single-stranded RNA virus in the family *Paramyxoviridae* [12]. The genome is either 15,186, 15,192, or 15,198 nucleotides long [13,14,15,16] and contains six transcription units encoding nucleocapsid (N), phosphoprotein (P, V, and W), matrix (M), fusion (F), hemagglutinin-neuraminidase (HN), and large polymerase (L) proteins [17]. The L protein is an RNA-dependent RNA polymerase that associates with the N and P proteins to form the viral replication complex [18]. This complex is responsible for transcription and replication of the viral genome. The F and HN proteins are involved in virus attachment and entry. The M protein is involved in the virus assembly process. Two additional proteins, V and W, are produced via RNA editing of the P gene [19]. The V protein is an interferon antagonist [20]. The function of the W protein is not established. The viral genes are ordered 3’-N-P/V/W-M-F-HN-L-5’ and are separated by short intergenic regions. The genes are flanked at the 3’ and 5’ ends by extragenic regions called the leader and trailer, respectively. The leader and trailer regions are involved in transcription and replication of the viral genome. The beginning and end of each gene contains conserved transcription signals known as the gene-start (GS) and gene-end (GE), respectively [18].

NDV strains have been classified into three pathotypes based on the severity of disease produced in chickens: low virulence (lentogenic), moderately virulent (mesogenic), and highly virulent (velogenic). Lentogenic strains cause mild or subclinical infection and are often used as live vaccines in the poultry industry. A major determinant of NDV virulence is the cleavability of the F protein into F1 and F2 subunits, which is necessary for virus infectivity. In velogenic and mesogenic strains, the F protein cleavage site contains multiple basic residues, which is cleaved by ubiquitous intracellular furin-like proteases. The F cleavage site of lentogenic strains contains fewer basic residues that cannot be cleaved by furin-like proteases but is cleaved by trypsin-like secretory proteases found only in the respiratory and enteric tracts, which restricts their replication to those sites [17].

## 4. Advantages of NDV as a Vaccine Vector for SARS-CoV-2

NDV has several advantages over other viral vectors for the development of a vector vaccine against SARS-CoV-2. The effectiveness of NDV-vectored vaccines has already been evaluated against SARS-CoV in monkeys [8], against MERS-CoV in camels [9], and against avian infectious bronchitis virus (IBV) in chickens, a natural host challenge model [10].

### 4.1. Safety

#### 4.1.1. Safety of NDV Strains

The lentogenic NDV strains such as LaSota or B1 used as a vaccine vector are highly attenuated and safe in poultry. These strains have been used for more than 70 years as live virus vaccines in the poultry industry with a good track record of safety and efficacy [17]. This feature makes lentogenic NDV strains unique among all live viral vaccines. In contrast, other currently used live viral vaccines in humans and animals are not naturally attenuated. Therefore, these NDV strains are not likely to cause disease in any wild or domestic avian species. Studies have shown that insertion of foreign genes into the genomes of NDV results in reduced rather than increased pathogenicity in birds. Therefore, the vaccine construct of NDV containing the S gene of SARS-CoV-2 will not pose any environmental or agricultural risk.

#### 4.1.2. Preclinical Studies in Monkeys

NDV is an avian virus and does not naturally infect non-avian species. It is highly attenuated in all non-avian species, including humans, due to natural host-range restriction. The safety and immunogenicity of NDV-vectored vaccines have been extensively studied in monkeys, a non-human primate model that is closely related phylogenetically and anatomically to humans, and which is a good model for pre-clinical studies of human vaccines [8,21,22,23,24,25]. In one study, intranasal and intratracheal inoculation of African green and rhesus monkeys with 1 × 10^6.5^ plaque forming units (PFU) per site with either a lentogenic strain (LaSota) or a mesogenic strain (Beaudette C) of NDV did not cause any clinical disease signs, and there was little or no virus shedding in throat swabs or tracheal lavage. Analysis of necropsy specimens showed only low-level replication of the virus in the upper respiratory tract and lungs, and no virus was detected in the blood and other organs [25]. In another study, African green monkeys were immunized through the respiratory tract with two doses of 10^7^ PFU of NDV expressing the S glycoproteins of SARS-CoV. The vaccinated monkeys developed robust SARS-CoV-specific neutralizing antibodies. The immunized animals were challenged with a high dose of SARS-CoV, and tissue samples were collected at the peak of viral replication (day 4). Viral assay of the lung samples showed that there was a substantial reduction of challenge virus titer in the immunized animals compared with control animals [8]. These results suggest that NDV is highly attenuated in non-human primates and its replication is restricted only to the respiratory tract.

#### 4.1.3. NDV Infection in Humans

There have been reports of NDV infection in humans involving poultry farmers and laboratory researchers [26]. In rare cases, it causes disease, which is usually mild, with conjunctivitis, laryngitis, and flu-like symptoms. These symptoms are self-limiting and disappear in 1 to 2 days [27,28]. In a study in India, 38% of poultry workers were found to be seropositive for NDV compared with less than 4% of the general population, indicating that most NDV infections in poultry workers are asymptomatic [29]. In another study in the United States, 29% of poultry workers also were found to be seropositive for NDV [30], again showing that most NDV infections in poultry workers are asymptomatic [29]. Moreover, there is no evidence for human-to-human transmission.

#### 4.1.4. Human Clinical Trials with Direct NDV Administration

NDV has been investigated as an oncolytic agent due to its selective replication in tumor cells and associated cell death eliciting innate and adaptive antitumor immune responses [31]. Several clinical trials have been conducted utilizing various pathotypes of NDV in human cancer patients [25,26,27]. Direct inoculation of NDV into thousands of patients by intranasal or parenteral routes has resulted in only mild side effects. The common side effects have been mild flu-like symptoms, conjunctivitis, and laryngitis [32,33]. In a dose-escalation trial, cancer patients were administered one or more increasing doses of a mesogenic NDV strain. The NDV strain was well tolerated in doses of at least 3 × 10^9^ infectious units by the intravenous route and at least 4 × 10^12^ infectious units by the intratumoral route. The infected patients displayed mild flu-like symptoms and approximately 50% of the patients had transient low-level virus shedding [34]. In another clinical study, inhalation of approximately 1 × 10^8^ PFU of NDV was associated with only one day of low-grade fever in 25% of patients [32]. To date, there has been no report on accumulative toxicity associated with repeated vaccination with NDV [34]. Overall, the safety of NDV strains as anticancer agents has been consistently high with mild side effects, suggesting that NDV is highly safe in humans.

### 4.2. Immunogenicity

NDV strains mimic restricted natural infection in the respiratory tract of avian and non-avian host cells resulting in the inducement of innate and adaptive immunity. NDV potentially expresses the protective epitopes of the foreign protein in native conformation in host cells that is crucial in order for the S protein of SARS-CoV-2 to induce a protective immune response. It also induces good local immunity in the respiratory tract where pathogens like SARS-CoV-2 enter the body. The restricted replication of NDV in the respiratory tract of humans would lead to the induction of a balanced immune response. Development of a balanced immune response is less likely to lead to the antibody-dependent enhancement of disease, which is sometimes associated with vaccination. NDV vector vaccines have shown promising results against SARS-CoV and IBV [8,10].

NDV possesses strong immunostimulatory properties through the induction of large amounts of type1 interferon (IFN) and by its inability to block type1 IFN response in human cells.

The viral surface proteins (F and HN) also play an important role for the immunostimulatory properties of the virus by upregulating major histocompatibility complex (MHC) and cell-adhesion molecules and facilitating lymphocytes and antigen-presenting cells through their expression at the surface of infected cells. [35].

NDV encodes only six major proteins. Therefore, the antigenic competition between the foreign protein and the vector proteins is much less than other DNA viral vectors, whose genomes encode large numbers of proteins.

NDV is an avian virus, which avoids the problem of pre-existing immunity to the vector. This is a potential problem with viral vectors of common human viruses or viral vectors that are antigenically related to common human viruses. Serological studies have indicated that more than 96% of the human population is seronegative for NDV [26]. Therefore, the entire human population is amenable to NDV-vectored vaccines.

### 4.3. Genetic Stability

Foreign genes in NDV are stable. NDV does not undergo the genetic recombination or genetic reassortment observed for certain RNA viruses. NDV is an acute cytoplasmic RNA virus, which precludes concerns of long-term infection or integration into host cell DNA. Modular organization of the NDV genome facilitates the insertion of foreign genes.

## 5. Construction of an NDV-Vectored Vaccine against SARS-CoV-2

Infectious NDV can be produced from cloned cDNAs by a method called reverse genetics. This involves co-transfecting cells with four T7 polymerase driven plasmids. One plasmid encodes a positive-sense copy of the genome (positive-sense is used instead of negative-sense to avoid hybridization of the plasmid expressed RNA with viral mRNAs) and three plasmids encode the N, P, and L proteins of the viral polymerase complex. [36,37]. The T7 RNA polymerase inside the cell is provided either by a recombinant vaccinia virus, from a co-transfected T7 expression plasmid, or a cell line that expresses T7 polymerase. This creates an artificial viral replication cycle inside the cell and leads to the production of infectious NDV. This method allows for insertion of a foreign gene into the viral genome.

To construct the NDV-vectored SARS-CoV-2 vaccine, the open reading frame encoding the S protein of SARS-CoV-2, the codon optimized for higher expression in human cells, will be engineered to contain the NDV-specific gene-start and gene-end sequences at the beginning and end of the ORF, respectively (Figure 1). The S gene transcription cassette is then inserted into a 3’non-coding region of an NDV gene as an additional transcription unit [8,10]. Owing to polar gradient transcription, foreign genes are expressed more efficiently when placed closer to the 3’-end of the genome [17]. A foreign gene can be placed between any two genes of NDV, but the insertion site between the P and M genes has been found to be optimal for efficient expression of the foreign gene replication of NDV [38,39,40,41]. NDV can accommodate foreign sequences of at least 5.0 kb with a good degree of stability [17]. Therefore, NDV will not have any problem in accommodating the S gene of SARS-CoV-2, which is approximately 3.8 kb in length.

## 6. Conclusions

There is a dire need for the development of an effective vaccine against SARS-CoV-2. However, the development of a safe and effective vaccine against SARS-COV-2 is a challenging task. Prior studies from other pathogenic coronaviruses of humans (SARS and MERS) have indicated the potential of NDV as a vaccine vector. NDV has several features that merit attention with respect to vaccine safety and efficacy. Excellent data exist on the safety of NDV in humans that establish and monitor safety, and also define correlates and the durability of protection. Development of an NDV-vectored vaccine may result in an effective countermeasure to curtail the present pandemic and any future pandemics of SARS-CoV-2 infection.

## Figures and Tables

**Figure 1 pathogens-09-00619-f001:**
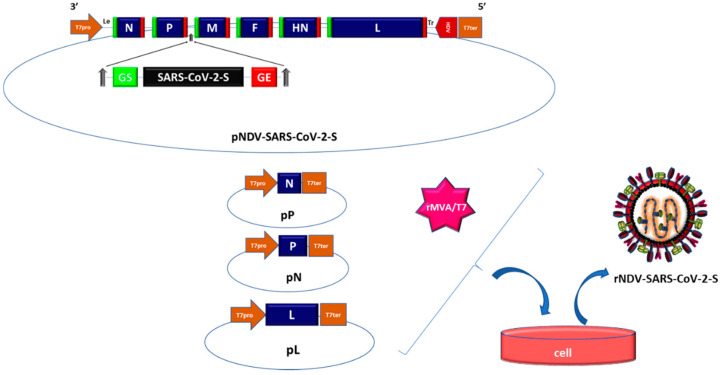
Schematic diagram for generation of rNDV expressing S protein of SARS-CoV-2-S protein. Infectious Newcastle disease virus (NDV) can be produced from cloned cDNAs by a method called reverse genetics. Infectious rNDV expressing SARS-CoV-2 is generated by co-transfecting cells by four T7 polymerase driven plasmids, which include one plasmid encoding a positive-sense copy of the genome of NDV containing the sequence of the S gene of SARS-CoV-2 flanked between P and M genes of NDV, and three plasmids encoding the N, P, and L genes, and a recombinant vaccinia virus expressing T7 polymerase.

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
