# Peer review of "Newcastle Disease Virus as a Vaccine Vector for SARS-CoV-2"

_pathogens, 2020, doi:10.3390/pathogens9080619_

Round 1
Reviewer 1 Report
Shrivani and Samal have prepared a review on the potential use of NDV as a vector for developing a vaccine against SARS-CoV-2, the virus causing the ongoing COVID-19 pandemic. The authors have reviewed studies that show the potential of NDV to serve as vector for vaccines against SARS-CoV and MERS-CoV. The authors have also summarized studies on NDV safety, previous studies in non-human primates, infections in humans, clinical trial, NDV immunogenicity, and genetic stability of the constructs.
English/style/grammar is sound, only minor polishing is necessary.
Use seminal papers, there are several occurrences where non-seminal papers have been used - e.g. genome organisation.
Lines 8-9: update numbers
Line 14: well-suited properties
Line 20: put abbreviation of COVID-19 in parenthesis on first occurrence.
Line 21: update numbers
Lines 31-32: provide S-gene size
Line 47: which pathogen?
Lines 45-51: The passage assumes that an NDV-SARS-CoV-2 vaccine will have these characteristics while there is no evidence to support these. Please re-phrase for clarity and that this has yet to be demonstrated.
Line 61: sentence seems incomplete
Line 76: provide all NDV sizes
Lines 78-79: the L-protein is the polymerase. While N and P have roles in replication and transcription, it is incorrect to say that "N, P and L are the viral polymerase"
Line 80: W has not been defined. There are 2 Ws in the gene order.
Lines 132-135: Where? It sounds this is valid for the whole world. Here and elsewhere - specify where the studies have been conducted, how the samples where collected, were the samples representative, proper statistics.
Line 152: which host?
Line 152-156: similar comment to above - any evidence of that the S-protein of SARS-CoV-2 will be expressed in native conformation? Re-word that this is potentially and needs to be studied and confirmed.
Lines 155-156: there is no evidence that the vaccine will be protective - reword
Lines 187-191: these statements need to be supported by at least a couple of references.
Lines 192-193: fix style
Author Response
Dear Reviewer,
Thank you for reviewing our manuscript entitled " Newcastle disease virus as a vaccine vector for SARS-CoV-2. We submitted the revised manuscript and a response to your comments. Thank you.
Sincerely,
Siba K Samal

Reviewer 2 Report
The authors made a good analysis on the suitability of exploring NDV as a potential vector vaccine against SARS-CoV-2. It is totally acceptable and valuable for publication in its present form as a mini-review.
Author Response
Dear Reviewer,
Thank you very much for reviewing our manuscript entitled " Newcastle disease virus as a vaccine vector for SARS-CoV-2". We have submitted the revised manuscript. Thank you.
Best regards
Siba K Samal